# Application of NF Polymeric Membranes for Removal of Multicomponent Heat-Stable Salts (HSS) Ions from Methyl Diethanolamine (MDEA) Solutions

**DOI:** 10.3390/molecules25214911

**Published:** 2020-10-23

**Authors:** Asma Ghorbani, Behrouz Bayati, Teresa Poerio, Pietro Argurio, Tavan Kikhavani, Marzieh Namdari, Licínio M. Ferreira

**Affiliations:** 1Department of Chemical Engineering, Ilam University, Ilam 69315-516, Iran; asmaghorbani94@gmail.com (A.G.); t.kikhavani@ilam.ac.ir (T.K.); 2National Research Council of Italy, Institute on Membrane Technology (CNR-ITM), 87036 Rende (CS), Italy; 3Department of Environmental Engineering, University of Calabria, I-87036 Rende (CS), Italy; pietro.argurio@unical.it; 4Department of Chemical Engineering, Iran University of Science and Technology, Tehran 16846-13114, Iran; mhs.namdari@gmail.com; 5Department of Chemical Engineering, Faculty of Sciences and Technology, University of Coimbra, 3030-790 Coimbra, Portugal; lferreira@eq.uc.pt

**Keywords:** amine purification, negative surface membrane, heat stable salts removal, amine rejection, nanofiltration membrane

## Abstract

This study presents an efficient and scalable process for removing the heat-stable salts (HSS) ions from amine solution while recovering methyl diethanolamine (MDEA) solution for its reuse in gas sweetening plants. The presence of HSS in the amine solution causes the loss of solvent capacity, foaming, fouling, and corrosion in gas sweetening units so their removal is crucial for a more well-performing process. Furthermore, the recovery of the amine solution can make the sweetening step a more sustainable process. In this study, for the first time, the removal of a multicomponent mixture of HSS from MDEA solution was investigated via a nanofiltration process using flat-sheet NF-3 membranes. The impact of operating parameters on salts and amine rejection, and flux, including the operating pressure, HSS ions concentration, and MDEA concentration in the feed solution was investigated. Results based on the nanofiltration of an amine stream with the same composition (45 wt.% MDEA solution) as that circulating in a local gas refinery (Ilam Gas refinery), demonstrated a removal efficiency of HSS ions in the range from 75 to 80% and a MDEA rejection of 0% indicating the possibility of reusing this stream in the new step of gas sweetening.

## 1. Introduction

Oil and gas companies need to improve their energy efficiency due to the high increase in demand and overall energy costs [1]. The main sources of energy production in the industries are fossil fuels, including natural gas, coal, and crude oil which are nonrenewable. Among these sources, natural gas has greater advantages than other fossil fuels. For instance, compared to other fossil fuels, it is more environmentally friendly [2]. Furthermore, it has combustible properties and is widely used in the power generation and petrochemical industries [3]. In general, natural gas contains methane (75–90%), propane, butane, ethane, heavy hydrocarbons (1–3%), and also impurities such as water vapor, carbon dioxide (CO_2_), hydrogen sulfide (H_2_S), mercaptans (RSH), carbonyl sulfide (COS), carbon disulfide (CS_2_), and nitrogen [4,5,6]. Some of these impurities are toxic, corrosive, and cause environmental pollution after combustion so their removal from natural gas is extremely important [7,8]. One of the techniques for treating the natural gas stream is to use an aqueous alkanolamine such as methyl-diethanolamine (MDEA) which is prevalent in the removal of up to 30% of impurities from the gas stream [8,9,10,11].

MDEA is a tertiary alkanolamine that is used to remove contaminants in the gas sweetening industries [12]. Using MDEA solution leads to the formation of MDEAH^+^ and other by-products such as SCN^−^, HCOO^−^, H_3_CCOO^−^, and H_3_CH_2_CCOO^−^. These by-products together with MDEAH^+^ ion form a heat-stable salts (HSS) system, which cannot be regenerated under convectional regeneration and therefore remain in the absorption solution [13]. The accumulation of the HSS causes various issues including foaming, corrosion, frequent filter replacement, high solution viscosity, and fouling [14,15]. Thus, HSS have significant negative effects on the efficiency of the removal of CO_2_ and H_2_S from the gas stream. Therefore, it is crucial to remove them from the amine solution and increase the free amine concentration level by a continuous absorption/regeneration process. Various technologies have been used to remove HSS ions from the amine solution such as thermal reclamation, distillation, ion exchange resins, electrodialysis (ED), and nanofiltration (NF) [16,17,18,19].

Pal et al. [20] investigated the removal of HSS anions from lean MDEA solvent using four commercially available anion exchange A-D resins. The results showed that resin C exhibited the best removal potential for HSS. Cho et al. [16] studied the enhancement of the low regeneration efficiency of HSS-loaded anion exchange resin by using a zirconium pentahydroxide displacement technique. They found that in the batch system, efficiency is 15.2% higher than conventional NaOH. While in the continuous mode, the efficiency is about 28.0–17.8% higher for 1.5–5 bed volume. Ion exchange is beneficial if the ions accumulation rate and HSS ions concentration are low in the amine solutions. Higher HSS concentration and total solution flow rates lead to higher equipment costs and scale which are not desirable [17,19]. Besides, ion exchange resin beds need to be regenerated from the sodium hydroxide solution, producing a lot of wastes with many issues from the environmental point of view.

As mentioned above, the ED process is one of the methods for removing HSS from the amine solution. Numerous studies have been conducted using ED to remove HSS from the amine solution. They found that ED is useful in removing anions and cations, but it can also cause absorbent losses during the process [21]. Bazhenov et al. [22] studied the removal of HSS during the reclaiming of degraded monoethanolamine (MEA) solutions by ED in a pilot plant. They found that ED provides quite a uniform removal of HSS ions regardless of the nature of anions, but ED is unable to remove heavy metal. In another study, lab-scale ED performance was evaluated by Volkov et al. [23]. They found that 70% of HSS were removed from MEA solvent in 30 min and the specific energy consumption was 7 Wh/g HSS. Moreover, Meng et al. [13] studied the removal of HSS ions from MDEA solutions by the ED process. Their results showed that about 90% of HSS were removed from aqueous MDEA solutions and the loss ratio was less than 9% in the best conditions. The removal of HSS ions from spent amine wastewater using an ED stack has been studied by Wang et al. [24]. Their results showed that ED performance was dependent on the applied current density. To overcome the electrical resistance, applying high current density is not appropriate due to higher energy consumption. However, the use of the ED process has some disadvantages including higher power consumption and more waste generation. Besides, the high costs associated with the manufacture of membranes resistant to high pH and amine solvents limit the use of ED systems [17,19].

The nanofiltration process is a simple and very efficient way of separating ions from water solutions. Nanofiltration, which is a relatively new pressure-driven filtration process with separation properties between reverse osmosis (RO) and ultrafiltration (UF) processes, has several advantages including higher permeability for monovalent ions, low permeability for multivalent ions, lower operating pressure, higher flux, lower energy consumption compared to RO, and lower molecular weight cut-off (200–1000 Da) compared to UF [25,26,27,28,29,30,31]. The mass transport process in NF involves a combination of different mechanisms including steric exclusion (sieving), Donnan (charge effects) interactions, dielectric exclusion, and solute-membrane affinity [32,33,34,35,36,37,38,39,40].

Due to the presence of amine in the adsorption solution (20–45 wt.% amine in water), the separation of stable salts from the solution is different from ion removal from water. Therefore, it is necessary to study the removal of HSS ions from the amine solution using a simple and economic NF process. Limited studies have been presented in the literature regarding the removal of HSS ions from the MEA solution using the NF process. Lim et al. [41] evaluated the use of NF membrane (Koch MPF-34 and MPF-36) to concentrate single salt of HSS ions from 30 wt.% MEA solution before reclamation. They found that NF can reject more than 80% of HSS ions, while the rejection of the MEA is less than 7%. Furthermore, in another work, we [42] investigated the performance of the NF membrane and transport coefficients for a mixture of binary salts (acetate and sulfate) in the MDEA solution using two models, including the Spiegler–Kedem–Katchalsky (SKK) model, and film theory and extended Nernst–Planck (FT-ENP) model. We found that there was an acceptable agreement between the experimental and calculated rejection curves for the two models.

The CO_2_ loading capacity of tertiary amines such as MDEA (1.0 mol CO_2_/mol amine) is higher than primary and secondary amines (0.5 mol CO_2_/mol amine) and stripping of CO_2_ from DEA or MEA during the regeneration process requires more energy than MDEA. Furthermore, the primary and secondary amines (MEA and DEA) are more susceptible to chemical instability and corrosiveness than tertiary amines such as MDEA [43]. Thus, considering the importance of the issue, in the present study, the removal of HSS ions from MDEA solution using a nanofiltration membrane is investigated. To our knowledge, this is the first study on the removal of a multi-component mixture of HSS from MDEA solution using the NF membrane. For this purpose, the NF membrane was supplied and then characterized by zeta potential measurements and Fourier-transform infrared spectroscopy (FTIR) analyses. Besides, the effect of operating pressure on the ions rejection and the permeate flux at different MDEA solution concentrations was studied. Furthermore, the effect of MDEA concentration on HSS ions rejection, MDEA rejection, and permeate flux was investigated at an operating pressure of 70 bar. Moreover, the effect of concentration of HSS ions in 45 wt.% MDEA solution on the permeate concentration and the ions rejection was also evaluated at 70 bar. The final aim of the present study was to find an efficient and scalable process for removing the HSS ions from amine solution, to reduce viscosity and corrosivity while recovering MDEA solution to be reused in gas sweetening plants.

## 2. Results and Discussion 

### 2.1. Membrane Characterization

FTIR analyses of nanofiltration membranes before and after their soaking for two months in a 45 wt.% of MDEA solution are shown in Figure 1. The spectrum of membrane before use (a) evidenced, as expected, a peak of 3433 cm^−1^ indicating the amide N-H vibration stretching and another peak at 1634 cm^−1^ indicating the presence of amide carbonyl (C=O) bond. Moreover, the peaks at 874 cm^−1^, 852 cm^−1^, 793 cm^−1^, and 726 cm^−1^ represent the polysulfone structure with aromatic C-H bending, and the peak at 1729 cm^−1^ is related to the ester groups of polyester [33]. In addition, the overlap of stretching band of N–H (amide groups) and that of carboxylic acid groups (O-H stretching) from the imperfect cross-linking of the polyamide layer lead to the creation of the wide band around 3433 cm^−1^ [44,45]. The spectrum of the membrane after its soaking for 2 months in 45 wt.% MDEA solution (b) evidenced an overlap of amine peaks (e.g., N-H bending at 1630 cm^−1^) due to the adsorption of amine on the membrane surface. The amide N-H vibration stretching with the peak at 3433 cm^−1^ indicated a good chemical stability of the membrane in contact for two consecutive months with a 45 wt.% MDEA solution.

The separation mechanisms in NF membranes are usually described by the charge and sieving effects. The membrane surface charge plays an essential role in determining the NF membrane separation properties [46]. For this purpose, the measured average of the surface zeta potential of the NF-3 membrane was −42 mV with ±0.54% error at a pH of 7.4. Therefore, the membrane has a negative surface charge at this pH. This result agrees with the one obtained by Lin et al. [47]. They measured the surface charge of the Sepro NF membrane by zeta potential by changing the pH from 2.5 to 10.6. The membrane has a positive charge at pH 2.5 with a zeta potential value of 31 mV, with an isoelectric point (IEP) at pH 5. The zeta potential at pH 10.6 was −72 mV, showing an extremely negative membrane surface charge at the high pH values. The negative charge of the membrane is due to the deprotonation of the carboxylic groups on the surfaces of the highly crosslinked polyamide skin layer membrane [47].

### 2.2. Membrane Fluxes and Osmotic Pressure

Nanofiltration membranes for use in separation systems must use an applied pressure that overcomes the system’s osmotic pressure, to obtain a flow of a solvent across the membrane from a higher solute concentration side to a lower concentration side. The solutes present in the amine solution are MDEA and a mixture of HSS, but the amine concentration (20–45 wt.%), is much higher than the concentration of the HSS ions; thus the osmotic pressure caused by these ions is negligible compared to the osmotic pressure due to amine concentration so the membrane flux is strongly dependent on MDEA concentration. The osmotic pressures for different MDEA concentrations are calculated and presented as an insert in Figure 2 and according to Equation (4), the osmotic pressure increases by increasing the amine concentration.

Figure 2 also shows the effects of operating pressure on permeate flux in the presence of the mixture of ions at different MDEA concentrations. It can be observed that at all amine concentrations, the permeate flux through the membrane increased by increasing the pressure up to 60 bar for 20 wt.% MDEA solution and up to 70 bar for 30 and 45 wt.% MDEA solution [42,48]. This linear trend can be explained considering that the driving force for the permeation through the membrane is the applied pressure gradient. Thus, the increase in the applied pressure increases the permeate flux. Besides, the linear trend of solvent flux observed in Figure 2 by increasing pressure till a certain value depending on MDEA concentration indicates a slight effect of concentration polarization phenomena operating at these pressures.

A deviation from the linear trend can be observed at high pressure (70 bar for 30 and 45 wt.% MDEA concentration, ca. 60 bar for 20% MDEA concentration). This indicates the presence of high concentration polarization on the membrane surface by operating at these high pressures, corresponding to high fluxes. Besides, a higher deviation from linearity was observed for a MDEA concentration of 45 wt.% due to the osmotic pressure increases which lead to a reduction in the permeate flux.

Generally, the operating pressure of the nanofiltration processes is 5~25 bar. However, the operating pressure in this study is higher than this range due to the high concentration of amine (20–45 wt.%) which increases the osmotic pressure of the solution.

### 2.3. Effect of Operating Pressure on the Ions Removal from MDEA Solution Containing Multicomponent Mixture of HSS

Figure 3a–c shows the results of HSS ions rejection from the MDEA solution (20–45 wt.%) by the NF3 membrane as a function of the operating pressure.

Results indicated that by increasing operating pressure, ions rejection for all MDEA solutions is quite constant. This trend is due to two counteracting mechanisms. The solvent flux increases by increasing the applied pressure (convective transport), favoring a concentration polarization increase due to the transport and accumulation of ions on the membrane surface leading to a reduction of the ions rejection [26,42,49]. On the other hand, ions flux is reduced due to steric and/or electrical hindrance and ions rejection is increased [50]. As a result, the ions rejection remained constant when increasing the applied pressure. This result agrees with the literature findings that indicate that the two phenomena (the solvent flux and concentration polarization) have opposite effects on the ions rejection [51,52].

Furthermore, the results evidenced a higher HSS ions rejection for 20 and 30 wt.% MDEA concentrations compared to 45 wt.% MDEA concentration.

It should be noted that the rejection of oxalate ions at all operating pressures and all MDEA solution concentrations was almost 100%. However, the rejection of acetate ion varied by changing pressure and the MDEA concentration. For 30 and 45 wt.% MDEA solutions, the rejection of acetate ions had fluctuation with increasing pressure (Figure 3b,c), which indicates that both concentration polarization and convective transport took place. However, all variations in rejection were small; hence they could be considered constant. The difference in rejection of the different HSS ions reported in Figure 3 can be explained by considering the interaction between the charge of ions and charge of membrane surface (the Donnan exclusion), as well as the difference between the size of the molecules and the membrane pore size (the steric exclusion). The charge repulsion between ions and the membrane surface led to an increase in rejection (Figure 4a). As shown in Figure 3a–c, the rejection of divalent anions (sulfate, thiosulfate, and oxalate) was greater than that of monovalent anions (acetate).

As mentioned before, size exclusion also plays an important role in the rejection of HSS ions. The observed difference between thiosulfate, sulfate, and/or oxalate rejection can be explained by the hydration energy. The hydration energy of different salts can be obtained from the following equation [48]:(1)$$E_{hyd}=148+556\left(\frac{q^{2}}{M^{1/3}}\right)+576{\left(\frac{q^{2}}{M^{1/3}}\right)}^{2}$$
where the hydration energy is related to the molecular weight (*M*) and the ion charge (*q*). The hydration energy and the ionic strength of ions are presented in Table 1. As it is shown, the divalent ions with higher hydration energy and ionic strength are more prone to be rejected than formate and acetate ions.

The amine solution used in the Ilam gas refinery’s sweetening unit contains 45 wt.% MDEA. As shown in Figure 3c, the rejection of bivalent HSS ions from the 45 wt.% MDEA solution by NF3 membrane was high, and 75–85% of total ions were removed, which indicated a good performance of NF-3 membrane in amine purification (Figure 4b).

### 2.4. Effect of MDEA Concentration on the Removal of HSS Ions

The removal of HSS ions and the permeate flux at different MDEA concentrations are summarized in Figure 5. The decrease of permeate flux and rejection of ions obtained by increasing MDEA concentrations can be explained considering the increase of the osmotic pressure. Based on Equation (6), the permeate flux decreases with increasing the osmotic pressure difference (∆*π*) at constant applied pressure (∆*P*). Figure 5b shows that the rejection of all ions decreases when increasing the concentration of MDEA. According to Equation (5), by increasing the MDEA concentration and thus decreasing the solvent flux, the contribution of diffusion transport increases, which leads to an increase in the transport of ions through the membrane and decreases in ions rejection.

### 2.5. MDEA Rejection

The MDEA concentration in the absorbent aqueous solution is an essential parameter in the gas sweetening processes. Therefore, the rejection of MDEA and the MDEA concentration in the permeate is of great importance in the purification of amine by the NF membrane process. The MDEA rejection was investigated at 60–80 bar. The results are reported in Table 2.

According to the results, the MDEA rejection decreased by increasing the MDEA concentration. The MDEA rejection reached zero at all pressures for a 45 wt.% MDEA solution. The zero amine rejection can be due to the difference between the molecular size of MDEA and the pore size of the membrane. Since the molecular weight cut-off of NF-3 membrane is between 250–300 g/mol and the molecular weight of the MDEA is 119.163 g/mol, MDEA can permeate through the membrane. At low amine concentration, water flux through the membrane pores is high due to its small size compared to the amine, and as a result, the amine concentration in permeate side is lower than that of the feed. Conversely, at high concentrations of amine (45% by weight of MDEA) the flow of the solvent across the membrane decreases (see Figure 2). This trend indicates that the contribution of the convection mechanism has decreased and therefore the contribution of the diffusive mechanism must be considered [53]. Furthermore, at higher MDEA concentration the presence of polarization phenomena can increase the MDEA concentration on the feed side membrane with a further increase in the diffusive transport of MDEA. As a consequence of these combined effects, MDEA concentration in the permeate became equal to its concentration in the retentate, with zero rejection.

### 2.6. Effect of Ions Mixture Concentration on the NF Membrane Performance 

Since the concentrations of acetate and formate ions in the amine solution are highest, the effect of feed concentration on the removal of binary mixtures of acetate and formate in the 45 wt.% MDEA solutions was examined. The results are represented in Figure 6. By increasing the concentration level of ions, the permeate concentration was increased and the rejection of ions was decreased. It can be related to the diffusion transport effect. Increasing the concentration level of ions can reduce the charge effects of the ions and the membrane surface charge due to the double-layer compression effect. It can reduce the electrostatic repulsion between the ions and the NF membrane surface [54]. Therefore, decreasing the membrane surface charge at high concentration results in the higher transport of anions and a decrease in the rejection rate of the ions.

## 3. Materials and Methods

### 3.1. Chemicals and Membranes

MDEA (methyl-diethanolamine (C_5_H_13_O_2_N)) was supplied by Ilam Gas Treating Co. Sodium acetate (C_2_H_3_NaO_2_), sodium sulfate (Na_2_SO_4_), sodium oxalate (Na_2_C_2_O_4_), sodium thiosulfate (Na_2_S_2_O_3_.5H_2_O), and formic acid (CH_2_O_2_) were purchased from Ghatran Shimi tajhiz, Iran. Commercially available NF-3 membranes (Sepro Co. USA) with a molecular weight cut-off 250–300 Da and negative surface charge were used in this study without further treatment. The selection of this cut-off is due to the size of the MDEA molecules (MW = 119.163 g/mol). The membrane properties are reported in Table 3.

Nanofiltration experiments were performed in the cross-flow filtration system schematized in Figure 7. In this system, a flat-sheet NF-3 nanofiltration membrane was used to remove the HSS from MDEA solutions. The membrane effective area was 0.0113 m^2^. The membrane was immersed in distilled water overnight before being placed in the NF module. First, the filtration system was operated under 20 bar pressure to ensure that the system was sealed and the membrane was free of defects. To keep the concentration and volume of feed constant, the concentrate and the permeate streams were recycled into the feed container. Permeate samples were taken at different applied pressures after reaching a steady state. A high concentration of amine (20–45 wt.%) increases osmotic pressure; therefore, NF experiments were carried out at high operating pressures in the range of 40–80 bar. The temperature was maintained at 35 ± 1 °C by placing a tubular cooling coil in the solution. The feed and retentate flow rates were controlled by a high-pressure pump and the inlet and outlet valves. In this study, based on the low effective area of the membrane, the average feeding flow rate and recovery ratio were maintained respectively at 0.693 L/h and 45% for all tests. After each experiment, the membrane was washed several times with distilled water at a pressure of 20 bar to reduce the membrane fouling and restore the initial flow rate. In particular, the membrane was washed every 3 days when the permeate flux was reduced by 42% of the initial one. The concentration of acetate, formate, sulfate, thiosulfate, and oxalate ions in the feed solutions was 200, 300, 150, 150, and 200 mg/L, respectively. The concentration of ions was selected based on their concentration in the circulating amine solution in a local gas refinery. The pH of the synthetic amine solution in the presence of ions mixture was 12, but since the circulating amine solution contains a small amount of CO_2_ and H_2_S, its pH was 10. Therefore, the pH of the synthetic amine solution was adjusted to 10 by adding an appropriate amount of HCl.

The concentration of ions in the amine solution was measured using the Ion Chromatography system equipped with separating column Metrosep A-Supp5-250. Permeate conductivity was measured by a conductivity meter (LT12042070, Taiwan). The concentration of MDEA was measured by the titration method with 1 M HCl and the chamber gas mixture reagent.

### 3.2. Evaluation of Membrane Performances and Governing Equations

The NF membrane performance was evaluated by calculating the ions rejection (*R_i_*) and permeate flux (*J_P_*) using Equations (2) and (3), respectively [55].
(2)$$\%R_{i}=\left(1-\frac{C_{P_{i}}}{C_{f_{i}}}\right)\times 100$$
(3)$$J_{P}=\frac{V}{t\cdot A}$$
where *R_i_* is the rejection of ions, *C_fi_* and *C_Pi_* are the concentration of ions in the feed solution and in permeate (mg/L), respectively, *J_P_* is the permeate flux (L/m^2^h), *V* is the permeate volume (L), t is the permeation time (h), and *A* is the membrane area (m^2^).

The osmotic pressure was evaluated according to Van’t Hoff equation (Equation (4)) [56]:(4)π=CRT
where *π* is osmotic pressure (kPa), *C* is the concentration of MDEA (mol/L), R is the universal gas constant, and T is the absolute temperature (K).

The solute transport through the membrane was explained using the Extended Nernst–Planck (ENP) equation, Equation (5) [57]:(5)$$J_{s_{i}}=K_{i,c}c_{i}V-K_{i,d}D_{i,\infty }\frac{dc_{i}}{dx}-z_{i}c_{i}K_{i,d}D_{i,\infty }\frac{F}{RT}\frac{d\Psi }{dx}$$
where *c_i_* and *D_i,_*_∞_ are the concentration of ith species within the pores and the molecular diffusion coefficient of species i at infinite dilution, respectively. *z_i_* is the valence of ion *i*, *K_i,c_* and *K_i,d_* are the convective and diffusive hindrance factors, respectively. *F* is the Faraday constant, *V* is the solvent velocity, and is the electrical potential gradient along the pore. The first term of Equation (5) is the convection term and it is related to the transport of solute with the solvent flow. The second term is the diffusion term which corresponds to the transport of species due to the concentration gradient. The last term is the electromigration term which refers to ion transport due to the gradient of electrical potential The transport equation has a series of constraints and boundary conditions such as the electroneutrality conditions of the feed solution, membrane and permeate respectively, and the Donnan steric-partitioning condition that relates the compositions in the membrane at the feed wall to the solute compositions at the membrane.

The solvent flux was calculated by using Equation (6) based on the Hagen–Poiseuille model [58]:(6)$$J_{V}=L_{P}\left(∆P-∆\pi \right)$$
where *J_V_* is the solvent flux, *L_P_* is solvent permeability coefficient, ∆*P* is the transmembrane pressure, and *J_V_* is the osmotic pressure difference between retentate and permeate. According to Equation (6), *J_V_* is reduced by increasing the ∆*π* at constant ∆*P*.

### 3.3. Membrane Characterization

The existence of functional groups on the membranes as well as their stability after use were evaluated by FTIR (FTIR, VERTEX 70, Bruker, Germany) spectrum analysis in the range of 400–4000 cm^−1^ wavelengths.

The electrical charge of the membrane surfaces and the isoelectric point were determined by measuring the zeta potential along the membrane surface using Electrokinetic SurPASS analysis (EKA Electro Kinetic Analyzer, Anton Paar, Austria). The measurements were carried out in a 1 mM KCl electrolyte solution at 22.5 °C at pH 7.37. The measurements were repeated four times, and the mean value was reported.

## 4. Conclusions

In this work, the removal of heat stable salts from 20–45 wt.% MDEA solutions by the nanofiltration membrane process was studied. The effects of operating pressure, HSS ions concentration, and MDEA concentration in the feed solution on the flux and the rejection were investigated. The NF membrane showed excellent performance for the removal of the HSS ions at lower MDEA concentration (20 and 30 wt.%). The permeate flux decreased with increasing the MDEA concentration from 20 wt.% to 45 wt.% due to the increment of osmotic pressure and the concentration polarization phenomena. By increasing the MDEA concentration at a pressure of 70 and 80 bar, the amine rejection through the membrane decreased until zero using a 45 wt.% the MDEA solution. The removal of HSS ions from 45 wt.% MDEA solution was about 75–80%, which indicates the acceptable performance of the NF-3 membrane.

The results based on the treatment of an amine stream with the same composition as that circulating in a local refinery (Ilam Gas refinery) indicated the possibility of obtaining a scalable process to remove HSS ions while recovering MDEA solution that can be reused in the sweetening gas units, obtaining relevant advantages in terms of sustainability of the process.

## Figures and Tables

**Figure 1 molecules-25-04911-f001:**
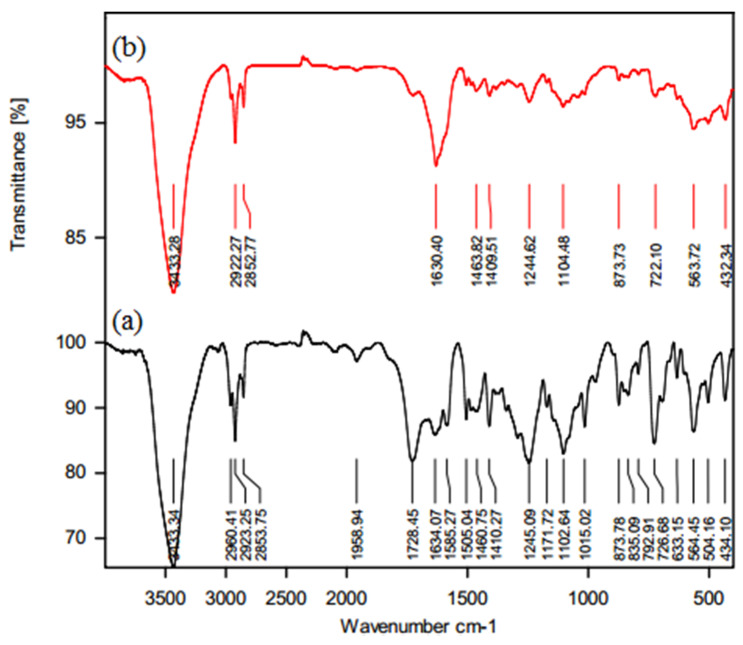
Fourier-transform infrared spectroscopy (FTIR) spectra of NF-3 membranes: before use (**a**); soaked for 2 months in a 45 wt.% methyl diethanolamine (MDEA) solution (**b**).

**Figure 2 molecules-25-04911-f002:**
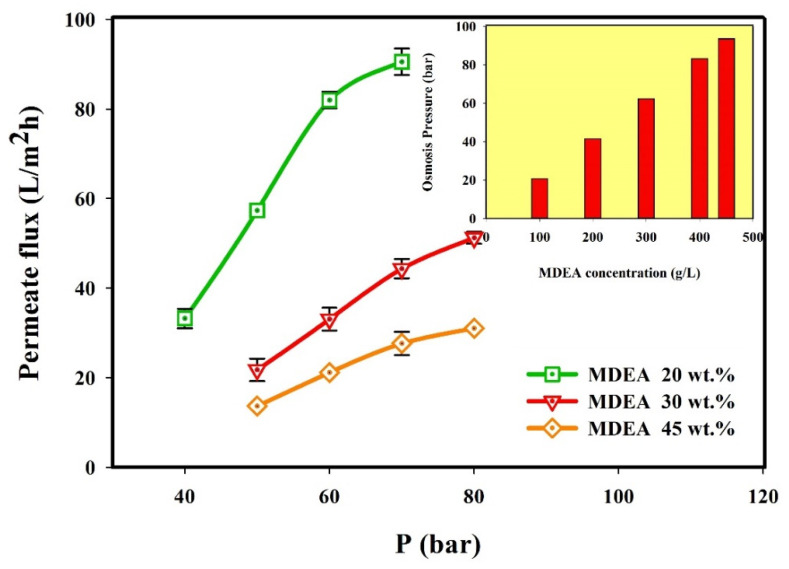
The effect of pressure on permeate flux for 20–45 wt.% MDEA solution in the presence of multi-component mixtures of ions by NF-3 membrane at 0.693 L/h, 35 °C and pH = 10; Insert: Osmotic pressure as a function of MDEA concentration.

**Figure 3 molecules-25-04911-f003:**
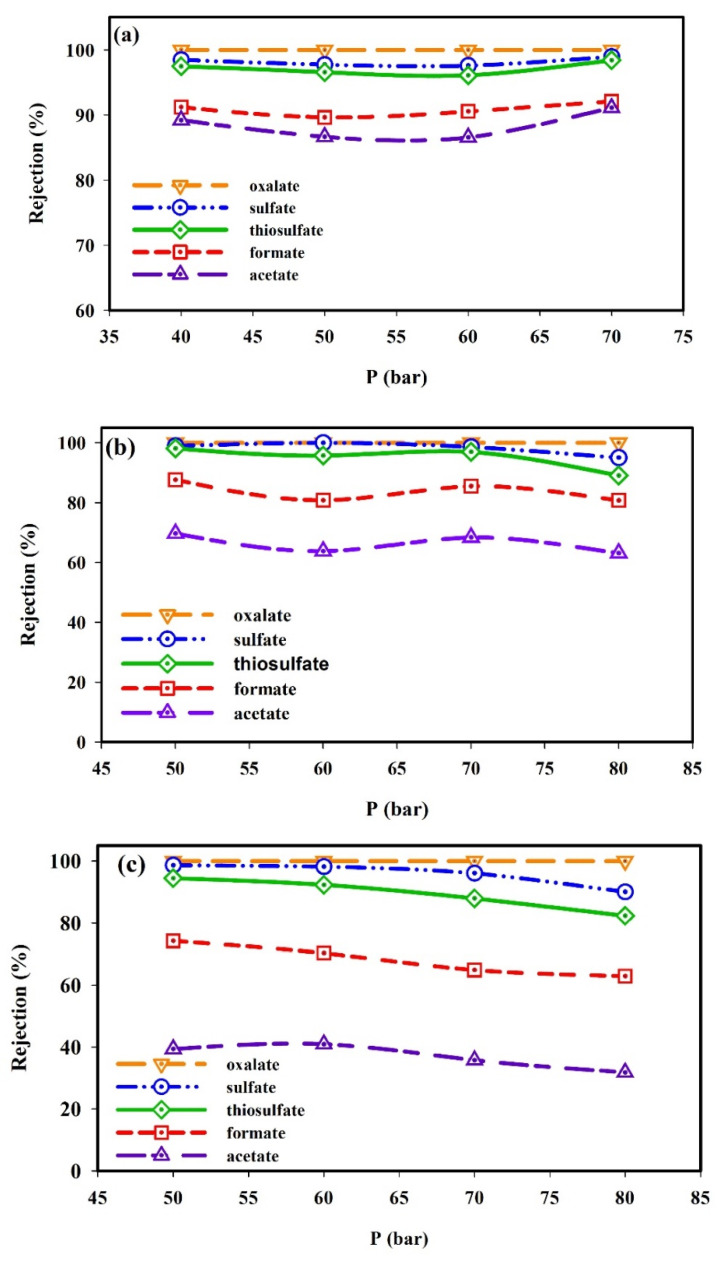
Effect of operating pressure on the rejection of multicomponent mixtures of ions in (**a**) 20 wt.%, (**b**) 30 wt.%, and (**c**) 45 wt.% of MDEA solution (temperature: 35 °C, pH = 10).

**Figure 4 molecules-25-04911-f004:**
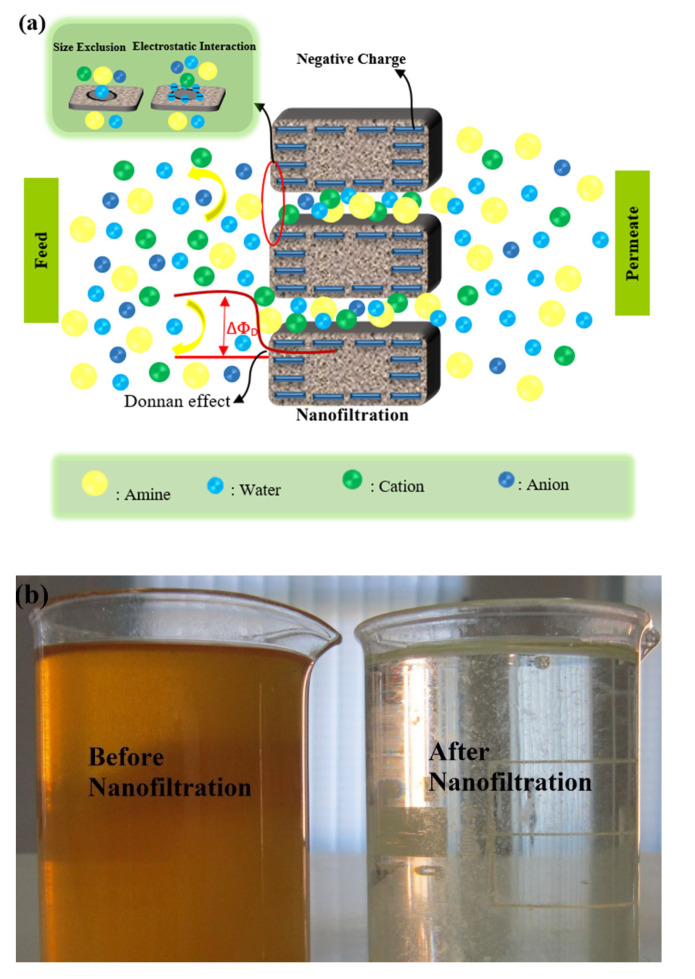
Schematic diagram of the mechanism of ions rejection performance from amine solvent by nanofiltration membrane (**a**), Samples of amine solution before and after nanofiltration process (**b**).

**Figure 5 molecules-25-04911-f005:**
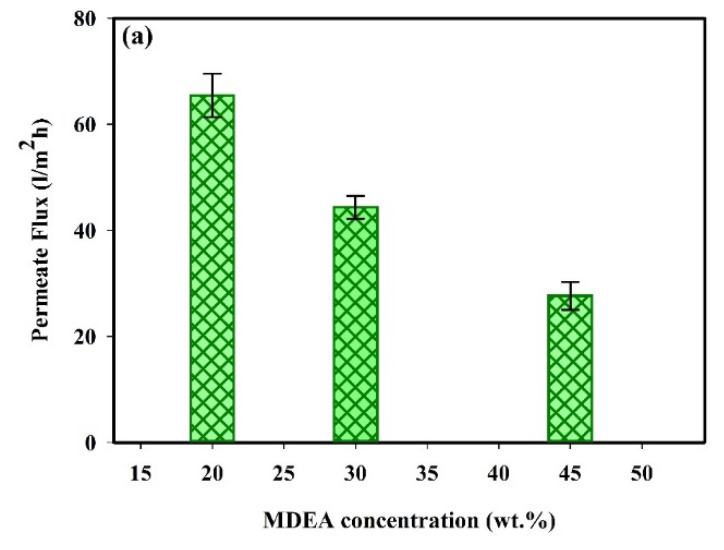
Effect of the MDEA concentration on (**a**) the permeate flux, and (**b**) the rejection of ions at the operation pressure 70 bar, 0.693 L/h, 35 °C and pH = 10.

**Figure 6 molecules-25-04911-f006:**
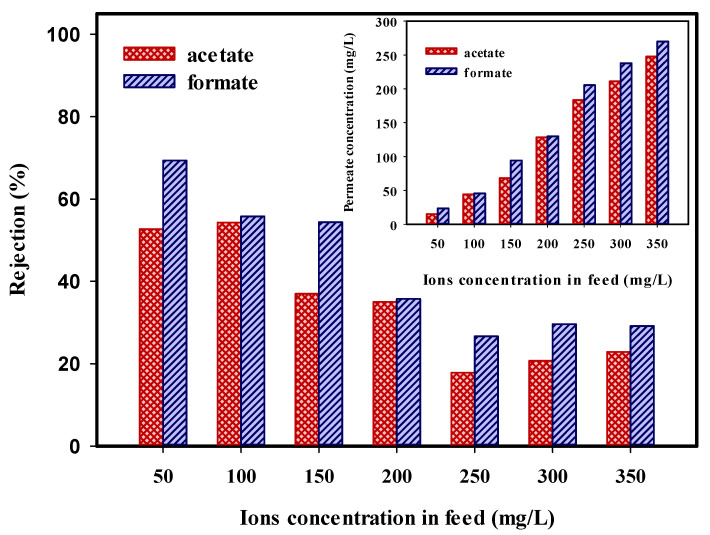
Effect of ions concentration in the feed on the rejection of the binary mixtures (acetate and formate) in the 45 wt.% MDEA solutions (Feed flow: 0.693 L/h; Pressure: 70 bar; Temp: 35 °C; pH: 10). Insert: Effect of ions concentration in the feed on the permeate concentration.

**Figure 7 molecules-25-04911-f007:**
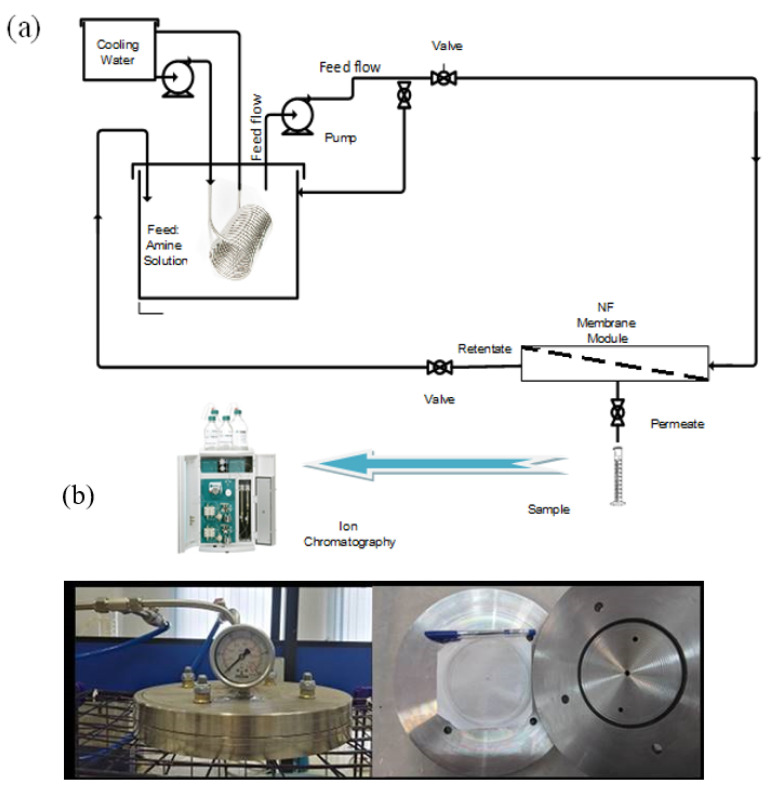
Scheme of experimental set-up (**a**) and images of nanofiltration (NF) membrane holder (**b**).

**Table 1 molecules-25-04911-t001:** Charge, molecular weight, hydration radius and hydration energy of ions used.

Ions	Molecular Weight (g/mol)	Hydration Radius (nm) (Kelewou, et al., 2011)	*E_hdy_* (kJ/mol)	Ionic Strength
C_2_H_3_O_2_^−^	59.04	0.260	328.94	0.00346
HCO_2_^−^	46.02	0.240	347.95	0.0065
C_2_O_4_^2−^	88.02	0.298	1113.05	0.00928
SO_4_^2−^	96.06	0.306	1072.84	0.00637
S_2_O_3_^2−^	112.13	0.323	1004.77	0.00547

**Table 2 molecules-25-04911-t002:** The rejection of MDEA from MDEA solutions at different pressures.

MDEA Conc. (wt.%)	Rejection (%)		
	60 Bar	70 Bar	80 Bar
20	22	22	-
30	20	17	11
45	0	0	0

**Table 3 molecules-25-04911-t003:** Properties of NF-3 Nanofiltration Membrane supplied by the manufacturer.

Membrane	NF-3
Manufacturer	Sepro Co.
Type	Flat, polymeric
Support layer	polyester support with a polysulfone substrate
Surface layer	Polyamide
Solute rejection ^a^ (%)	60% NaCl
	98% MgSO_4_
Operation limits	50 °C, 83 bar and 3–10 pH
Water flux (Lmh)	42
Molecular weight cut-off (Da)	250–300
Pore size (nm)	0.55

^a^ Test pressure = 10.3 bar and temperature = 25 °C.

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
