# Peer review of "Application of NF Polymeric Membranes for Removal of Multicomponent Heat-Stable Salts (HSS) Ions from Methyl Diethanolamine (MDEA) Solutions"

_molecules, 2020, doi:10.3390/molecules25214911_

Round 1

Reviewer 1 Report

attached

Author Response

Dear reviewer, a point-by-point reply to your comments can be found in the attached file.

The authors are grateful to the reviewer for helping to improve the quality of the work.

Reviewer 2 Report

Natural gas sweetening is important. Amine absorption method is currently the dominating method for the application; however, it is also limited by a number of challenges. The authors pursue to address one of them, heat-stable salts (HSS) removal, using NF membranes. This work is publishable if the authors can address the following concerns.

  1. Membranes are used to separate HSS from amine solution for natural gas sweetening, while had the authors considered to remove H2S and CO2 directly from natural gas using membranes? It will be more promising to completely replace the amine scrubber using membranes, which will be more energy efficient.
  2.  The authors propose to use a extremely high pressure, >70 bar, but this will cause a huge amount of extra energy input, capital cost etc. The authors may need to consider this aspect for the future application of their approach.
  3.  In Fig. 7, the membrane's permeate and retentate are illustrated wrongly

Author Response

(The authors gave the same response as above.)

Reviewer 3 Report

This article concerns with ion rejections containing amine solution through a polyamide NF membrane. The research field is suitable to the Molecules. The data and the discussions are original. However, there is a lack of specific description especially for the osmotic pressures or the concentration polarization. I think this article requires major revision for publication to the Molecules.

P4 line 160: The “osmotic pressure” should be specified. The extrapolation to the flux at 0 shown in the Figure 2 must be the same to the osmotic pressure.

P6 line 195: The concentration polarization should be calculated. It is hard to believe the following sentences without the calculation.

P7 line 209: In order to discuss the Donnan exclusion, the ion strength of the solution should be noted.

Figure 3: The stability of the membranes should be noted. Especially the data after the high pressure permeation tests are preferred.

Author Response

(The authors gave the same response as above.)

Round 2

Reviewer 3 Report

The revision is enough for publication in the Molecules.